# Influenza A virus reassortment is strain dependent

**Kishana Y. Taylor[1], Ilechukwu Agu[1], Ivy José[1], Sari Mäntynen[1], A. J. Campbell[1], Courtney Mattson[1], Tsui-Wen Chou[2], Bin Zhou[2¤], David Gresham[2], Elodie Ghedin[2,3], Samuel L. Díaz Muñoz[1,4]***

**1** Department of Microbiology and Molecular Genetics University of California, Davis Davis, California, **2** Center for Genomics and Systems Biology + Department of Biology New York University New York, United States of America, **3** Systems Genomics Section, Laboratory of Parasitic Diseases, National Institute of Allergy and Infectious Diseases, NIH, Bethesda, Maryland, United States of America, **4** Genome Center University of California, Davis Davis, California

¤ Current address: CDC, Atlanta, Georgia, United States of America
* samdiazmunoz@ucdavis.edu

**Data Availability Statement:** Raw sequence data are available in the NCBI Sequence Read Archive (SRX17867080). Code and data are available at the GitHub repository: https://github.com/sociovirology/human_influenza_LMGSeq.

## Abstract

RNA viruses can exchange genetic material during coinfection, an interaction that creates novel strains with implications for viral evolution and public health. Influenza A viral genetic exchange can occur when genome segments from distinct strains reassort in coinfected cells. Predicting potential genomic reassortment between influenza strains has been a long-standing goal. Experimental coinfection studies have shed light on factors that limit or promote reassortment. However, determining the reassortment potential between diverse Influenza A strains has remained elusive. To address this challenge, we developed a high throughput genotyping approach to quantify reassortment among a diverse panel of human influenza virus strains encompassing two pandemics (swine and avian origin), three specific epidemics, and both circulating human subtypes A/H1N1 and A/H3N2. We found that reassortment frequency (the proportion of reassortants generated) is an emergent property of specific pairs of strains where strain identity is a predictor of reassortment frequency. We detect little evidence that antigenic subtype drives reassortment as inter-subtype (H1N1xH3N2) and intrasubtype reassortment frequencies were, on average, similar. Instead, our data suggest that certain strains bias the reassortment frequency up or down, independently of the coinfecting partner. We observe that viral productivity is also an emergent property of coinfections, but uncorrelated to reassortment frequency; thus viral productivity is a separate factor affecting the total number of reassortants produced. Assortment of individual segments among progeny and pairwise segment combinations within progeny generally favored homologous combinations. These outcomes were not related to strain similarity or shared subtype but reassortment frequency was closely correlated to the proportion of both unique genotypes and of progeny with heterologous pairwise segment combinations. We provide experimental evidence that viral genetic exchange is potentially an individual social trait subject to natural selection, which implies the propensity for reassortment is not evenly shared among strains. This study highlights the need for research incorporating diverse strains to discover the traits that shift the

**Funding:** This project was funded by NIH grant 4R00AI119401-02 to SDM. SDM was supported by an NIH Pathway to Independence Fellowship (grant 1K99AI119401-01A1). KYT and IA were supported by NIH grant 4R00AI119401-02 to SDM. This work was supported in part by the Division of Intramural Research (DIR) of the NIAID/NIH (EG). EG received salary from NIH. SM was supported by the Academy of Finland Postdoctoral Fellowship (grant 1323426). The funders had no role in study design, data collection and analysis, decision to publish, or preparation of the manuscript.

**Competing interests:** We declare there are no financial, personal, or professional interests that could be construed to have influenced the work.

reassortment potential to realize the goal of predicting influenza virus evolution resulting from segment exchange.

## Author summary

Influenza A viruses are global pathogens that cause significant human morbidity and mortality through seasonal epidemics and pandemics that arise every few decades. These viruses have remained a public health threat in large part due to their high genetic variability, notably through exchange of their eight genome segments when two viruses coinfect a cell. Thus, determining the potential for genetic exchange has been an urgent goal, but has been difficult to achieve. Here we developed a novel method to quantify genetic exchange in a diverse collection of human influenza A viruses. We show that patterns of genetic exchange show strong strain-dependence, and do not necessarily track similarity between coinfecting strains or their subtype (H1N1 or H3N2). Our data support that specific strains can consistently bias genetic exchange, increasing or decreasing exchange independent of the other coinfecting strain. Our study suggests the propensity for reassortment is not evenly shared among strains and that identifying high-reassortment strains could improve pandemic preparedness or reveal new treatments.

## Introduction

RNA viruses can interact during coinfection of the same cell to exchange genetic material, creating novel strains that can evade immunity, and impact host range. Coinfection of influenza viruses provides the opportunity for their negative-sense RNA genome segments to be swapped during virion assembly, a process called reassortment [1–3]. Influenza A pandemics have been sparked by the emergence of reassortant strains [4], most recently with the 2009 H1N1 pandemic strain [5]. In the case of human seasonal influenza, every season there is a large collection of co-circulating lineages [6–9] that can reassort [7–10], creating novel variants that have, for instance, spread antiviral resistance to adamantane before its widespread use [11]. Aside from *intra*subtype reassortment in A/H1N1 and A/H3N2 strains that significantly shapes epidemiological dynamics within seasons [7–10], *inter*subtype reassortment occurs between these co-circulating lineages. Reassortment of H3N2 strains with both 2009 pandemic H1N1 and pre-pandemic H1N1 strains have resulted in the emergence [12] and *circulation* [13] of H1N2 strains in the human population. There is additional threat from frequent reassortment in strains that circulate in swine [14]–a "mixing vessel" for influenza due to the presence of α-2,3 and α-2,6 sialic acids in respiratory tissue–particularly considering the most common subtypes in swine (H1N1, H3N2, H1N2) overlap with circulating human strains [15] and that human strains have been repeatedly introduced into swine used in pork production [16,17]. Thus, reassortment has remained a topic of interest due to its substantial public health implications and consequences on the global epidemiology of influenza A viruses.

Predicting the outcomes of coinfection between intact strains (i.e. not experimentally manipulated, wild type) and potential reassortment is a longstanding goal of basic research and public health risk assessment. Experimental coinfection studies provide a powerful tool to assess coinfection outcomes and dissect patterns of segment exchange [1] and have resulted in substantial progress in identifying factors that limit or promote reassortment. The first experimental coinfection study [18] showed non-random association between segments in a A/PR/

8/1934(H1N1) and A/Hong Kong/1/68(H3N2) coinfection in which a lower-than-expected exchange among the three segments that form the polymerase complex was observed. Subsequent studies employing variations on the experimental coinfection approach supported non-random associations of segments between divergent heterosubtypic strains [19–21]. These results encouraged research into the factors that may restrict segment exchange [22] including: the replicative capacity of coinfecting viruses [23]; functional incompatibilities between proteins [21]; and RNA-RNA interactions involving genome packaging signals [19,22,24].

In contrast, some have argued that reassortment among intact strains was more pervasive than implied by these initial experimental coinfection studies [1,25–27]. Testing the conclusions of these studies to determine the reassortment potential of intact strains faced technical challenges: using reverse genetics could "force" any genotype combination but bypasses the infection process of intact viruses; allowing multi-cycle replication, which confounds the effects of between-cell competition on fitness with the intracellular process of reassortment; and genotyping a small sample of progeny viruses. These limitations were overcome by a landmark study [25] which showed that near-identical strains (differing only in synonymous substitutions to facilitate genotyping) of A/Panama/2007/99(H3N2) exhibited reassortment at levels approaching random expectations [25]. A second major contribution was tight control of infection conditions that limited infections to a single cycle, thus isolating reassortment outcomes from between-cell competition. This system was then leveraged to study reassortment of A/Panama/2007/99(H3N2) and A/Netherlands/602/2009(H1N1pdm), while controlling for homologous (self) reassortment. This study revealed that reassortment was efficient between strains of different subtypes, but that segment exchange was non-random, i.e., a bias towards specific segments or segment combinations (such as homologous PB2-PA combinations) was detected in the progeny. Thus, even though the *frequency* of reassortment may not differ between strains, the specific combinations of segments in the progeny can differ substantially. This result underscored that determining reassortment potential cannot be limited to a single measure. Several studies have incorporated an increasingly detailed accounting of reassortment potential [28,29] including: i) the proportion of progeny that are reassortant; ii) the number of genotypes produced; ii) the relative fitness of segments; and iii) the pairwise linkage of segments and other population genetic measures such as evenness and entropy. In sum, two conclusions regarding human influenza A have become widely accepted in the literature: First, that very similar strains generate a high proportion of reassortant progeny and a closer-to-random segment exchange, and, second, that divergence between parental viruses generates biased segment combinations.

Despite much progress, determining the reassortment potential between diverse influenza A strains has remained an elusive goal. Due to methodological limitations, studies to date have largely tested one strain pair at a time while studies involving more than two viral backgrounds are rare. Limited experimental studies have been done between diverse human seasonal strains spanning decades. This is crucial for three reasons. First, the factors that limit or favor reassortment can be biased by the selection of strains used to determine reassortment potential. Second, reassortment potential between strains may change as the virus evolves. Third, from an evolutionary perspective, while two viruses are required for genetic exchange, each may differ in their propensity to exchange genes [30,31], as observed in individual variability in meiotic recombination rate between animals [32–34]. More generally, in nature there is ample potential for diverse intact strains to encounter each other and reassort, underscoring the need to incorporate this variability into our experimental design [35].

To fill this gap, we developed a high throughput genotyping approach to facilitate quantification of influenza A virus reassortment between human strains. This tool allowed us to examine coinfection outcomes among a diverse collection of strains, representing multiple

epidemics; geographic origins; and both circulating human influenza A subtypes, H3N2 and H1N1. Our overall objectives were to test some of the conclusions in the literature regarding reassortment potential and address questions that remain intractable with current experimental approaches. Specifically, we aimed to: 1) use a statistically robust sample size to test strain traits (genetic similarity, subtype, epidemic season) that may influence reassortment potential; 2) provide a detailed accounting of coinfection outcomes (replication, proportion of reassortants, relative fitness of segments, segment linkage patterns, and genotype diversity); and 3) examine associations between coinfection outcomes (e.g. testing if viral replication of the coinfection is correlated with reassortment frequency).

We found a wide range of reassortment outcomes emerging from the pairwise strain combinations involved in the experimental coinfections. Of note, we find that high reassortment frequencies and random patterns of segment exchange were possible between divergent or heterosubtypic strains. Furthermore, there was no significant relationship between genetic similarity and various measures of reassortment. We show that viral kinetics vary independently of reassortment frequency, affecting the total number of reassortants produced by each coinfection. Most significantly, we found evidence of strain-dependent variation in reassortment wherein some strains bias reassortment frequency upwards, regardless of the coinfecting partner. This result suggests that reassortment potential may be strain specific and may be a trait of individual virus strains.

## Methods

### Cells and viruses

We maintained MDCK (Madin-Darby Canine Kidney) cells in minimum essential media supplemented with 5% FBS (Fetal Bovine Serum). Cells were checked for mycoplasma contamination by PCR. We obtained five wild type influenza A virus strains, a generous gift from the lab of Dr. Ted Ross, which we abbreviate here as: CA09, HK68, PAN99, SI86, TX12 (Table 1). These stocks were originally egg passaged and we made stocks through low multiplicity of infection (MOI) propagation in MDCK cells (obtained from ATCC/BEI); all strains were subjected to one passage. Each virus stock comprises a diverse population with mutations and internal deletions as is characteristic of natural [36] and lab [37] influenza viral stocks, even to some extent those created via reverse genetics [38].

### Infections and plaque assays

**Pairwise infections.** We conducted infections following the conditions of Marshall et al. [25]. Briefly, a 6-well plate with 1 X $10^6$ of MDCK cells was coinfected with a pair of strains in equal proportion at a MOI of 10. We synchronized infections at 4˚C for 1 hour to promote equal attachment of viruses. To limit infections beyond one cycle of replication, we conducted infections in the absence of trypsin [39,40] collected viral supernatants at 12hr post infection and stored them at -80˚C for subsequent genotyping and titration.

**Table 1. Influenza A strains used in experiments and their properties.**

| Strain Name | Subtype | Year | Region/Country | Abbreviation |
|---|---|---|---|---|
| A/Hong Kong/1/1968(H3N2) | H3N2 | 1968 | Hong Kong | HK68 |
| A/Singapore/1986(H1N1) | H1N1 | 1986 | Singapore | SI86 |
| A/Panama/2007/1999(H3N2) | H3N2 | 1999 | Panama | PAN99 |
| A/California/07/2009(H1N1)pdm | H1N1 | 2009 | California | CA09 |
| A/Texas/2012(H3N2) | H3N2 | 2012 | Texas | TX12 |

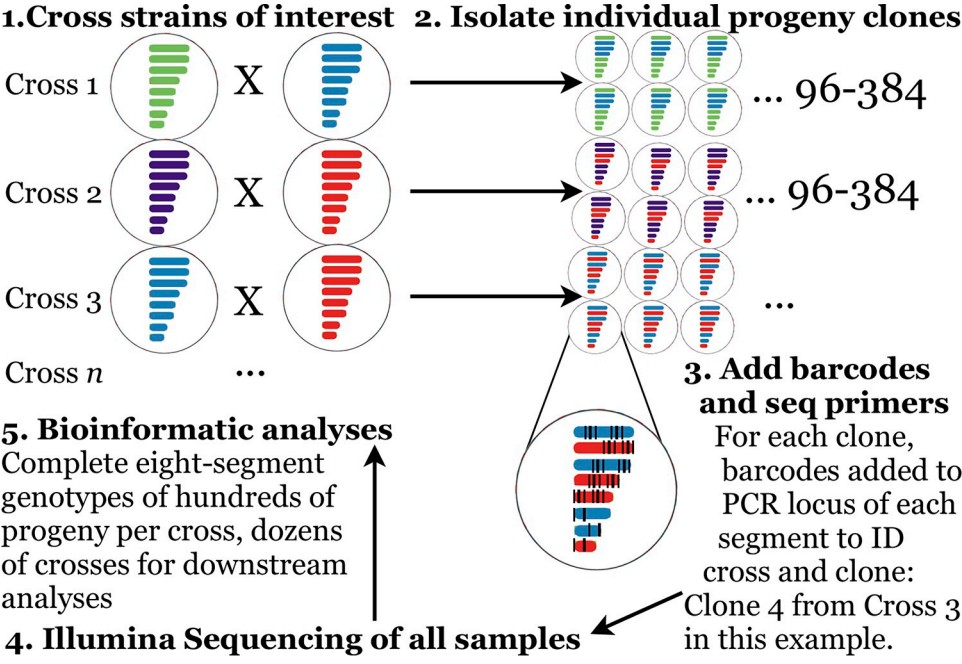

**Fig 1. Overall experimental workflow for experimental co- infections, progeny clone isolation, and genotyping.**

**Control infections.**   We conducted several control experiments to test the genotype by barcode sequencing approach (see below). First, we made biological replicates of the same coinfections for three strain pairs (HK68xCA09 n = 2, and HK68xPAN99 n = 2, and CA09/PAN99 n = 3). Second, we conducted infections of HK68xCA09 at increasing MOI's (0.01, 1, and 10), which are expected to lead to increasing levels of coinfection and reassortment [25]. Finally, we evaluated the parental stocks used to initiate coinfections to evaluate the genotyping pipeline and establish thresholds for assigning sequencing reads to specific strains.

To isolate clonal populations (i.e., individual, viable viral strains) for genotyping progeny from the infections, we conducted plaque assays. Plaques were picked with 1mL sterile pipette tips and placed in DPBS in deep-well 96-position plates for genotyping (see below). An overview of the steps in our experimental workflow are provided in Fig 1. We also conducted plaque assays from the same supernatants to quantify infectious particle production yield of experimental coinfection supernatants.

## Linked multilocus genotype by sequencing

To ascertain the eight-segment genotype of the viral progeny (i.e. isolated plaques representing clonal populations), we used a strategy we term **L**inked **M**ultilocus **G**enotype by **Seq**uencing (LMGSeq), inspired by Bar-seq experiments of yeast mutant libraries [41]. We describe the strategy in detail in Supplementary Material. Briefly, the picked plaques are used as RNA template in a one-step RT-PCR reaction with: 1) a uni13 primer (targeting a conserved region shared by all influenza A virus segment ends) barcoded to correspond with each of 96 wells; and 2) 10 primers (6 for internal segments and 4 primers customized for H1/H3 and N1/N2 subtypes), targeting the internal regions of genome segments barcoded to correspond with each coinfection (represented by a plate). These barcoded PCR products were pooled, diluted, and used as template in a second PCR reaction that added adapters for paired-end sequencing on the Illumina MiSeq platform. All LMGSeq experiments in this paper (including controls)

were conducted in a single MiSeq run. This protocol bears some similarity to other approaches [42,43], but was independently developed and varies in some experimental steps. In particular, our study design attaches barcodes in the first PCR, as opposed to adding them in the second PCR as in [42] or adding them twice as in [43], which then enables any amplicon to be pooled in a single PCR2 reaction. Consequently, the protocol moves from plate format to pooling and library preparation in a single tube earlier, decreasing library preparation costs exponentially and increasing throughput (see Supplementary Materials for details).

**Data analysis pipeline.** The analysis pipeline is posted on GitHub (https://github.com/sociovirology/human_influenza_LMGSeq). In brief, paired-end reads were demultiplexed and trimmed using CUTADAPT v2.6 [44], allowing only exact matches, resulting in files that contained reads for a single clonal population from a given coinfection. We conducted no further quality filtering, as quality scores were very high after read trimming and read merging using PEAR v0.9.6 [45]. We matched the reads of each file to a database containing the full-length amplicons for strains included in each cross, using the usearch_global command at 98% percent nucleotide identity in USEARCH v8.1.1861 [46]. The output of this database search was a file for each clonal progeny, which includes the match, PID, mismatches, and length of alignment, as well as other features, output for analysis in R (v3.6.3) [47].

**Genotype assignment.** To assign the full-length genotype of a progeny, each of the eight segments was assigned to one parental strain. Because progeny plaques were clonal and each segment was represented by multiple reads, we tallied the number of database matches and used a majority rule (i.e. consensus) to assign segments to a strain.

## Reassortment quantification and statistics

We scored individual progeny as reassortant if they contained one or more segments from different parental strains. We calculated reassortment frequency between two strains in a coinfection by calculating the proportion of reassortant progeny resulting from a given coinfection (proportion reassortant = reassortant plaques / total plaques isolated). We calculated deviations from random expectations in reassortment frequency using a Fisher exact test (eight segments in 256 unique combinations, 2 of which are parents; 254/256 = 99.22%). Because coinfections vary in the number of infectious progeny they generate (i.e. some coinfections are more productive in terms of viral replication), we estimated the *total* number of reassortants by determining the final titer of the supernatants (the same supernatant used to isolate plaques for genotyping) and multiplying by the respective reassortment frequency. We then set out to determine whether individual strains could influence reassortment rate of a coinfection, analogous to the general combining ability [48] on the recombination rate in animals [32–34]. While any given strain could have low or high reassortment frequencies, the strain could tend to generate a higher or lower reassortment frequency when controlling for the influence of the coinfecting partner. To determine whether particular strains had a tendency to increase or decrease reassortment, we used analysis of variance (ANOVA) to calculate the individual contribution of each strain to reassortment frequency.

## Quantification of genotypes generated by experimental coinfections

To estimate the unique genotypes produced by each experimental coinfection, we determined the eight-segment composition of each plaque and calculated genotype richness (S): the number of unique eight-segment combinations. To control for differences in the number of total plaques with complete eight-segment genotypes (N) isolated in each coinfection, we calculated the proportion of unique genotypes (S / N).

## Segment bias calculations

To calculate bias in the genotype segment composition according to its parental origin, we examined the proportion of parental origin in each individual segment (across a given experimental coinfection) and used a Fisher exact test to examine deviations from the null frequency of 0.50.

## Pairwise interactions among segments

We conducted pairwise analyses of segment associations and calculated linkage disequilibrium statistics to determine whether segment exchange was free among parental segments or whether segments were linked with respect to their parental origin. There are 28 possible segment pair combinations among eight segments. For each of these segment pair combinations, there are four potential combinations of the two segments derived from two parental strains: two of which are homologous and two heterologous. We calculated the proportion of plaque isolates that carried heterologous segment combinations for each for the 28 possible segment combinations, in each experimental coinfection.

## Effect of sampling on estimates of reassortment and segment bias

To determine how close experimental coinfections came to the 256 theoretically possible segment combinations, we conducted simulations to test the proportion of all possible segment combinations with the sample of 94 reads. Additionally, we used repeated experimental coinfections of select strain combinations to conduct simulations using empirical data to test the effect of sampling on the number of recovered genotypes.

# Results

## Genotype by barcode sequencing: a system for high-throughput quantification of reassortment between diverse strains

We used LMGSeq to ascertain the eight-segment genotypes of viral progeny derived from multiple experimental coinfections between our diverse panel of human influenza A strains. We sequenced 16 experimental coinfections (including control and replicate infections) in one MiSeq sequencing run resulting in a total of 25,729,649 reads, of which 15,501,869 were demultiplexed and yielded matches to the strain database. These reads covered 1,536 clonal progeny (mean reads / cross = 968,866 reads, range 219,326–1,544,248). The average read depth for each segment was 1,293x and was lowest for two polymerase complex segments (PB2 = 45.6x, PB1 = 16.3x), which are subject to large internal deletions, a well-known phenomenon [37,38]. These differences in coverage resulted in incomplete genotypes, primarily involving segment PB1. We established a majority rule for strain assignment (e.g. >50%), but on average strains were assigned at a higher proportion, $0.945 \pm 0.110$; overall, 90% of segments were assigned to a strain by over 75% of reads. Reassortment frequency estimates (i.e. proportion of genotyped progeny with segments from >1 parent) were repeatable among biological replicates in experimental coinfections, regardless of whether the coinfection generated low, medium, or high reassortment frequencies (**Fig 2A**). On average, replicate reassortment frequency estimates differed by ±0.06246 (range = 0.0100–0.1143). Reassortment frequency increased as a function of MOI and the data fit within the standard error of a model of exponential increase in reassortment as the MOI increased (**Fig 2B**), as previously reported [25].

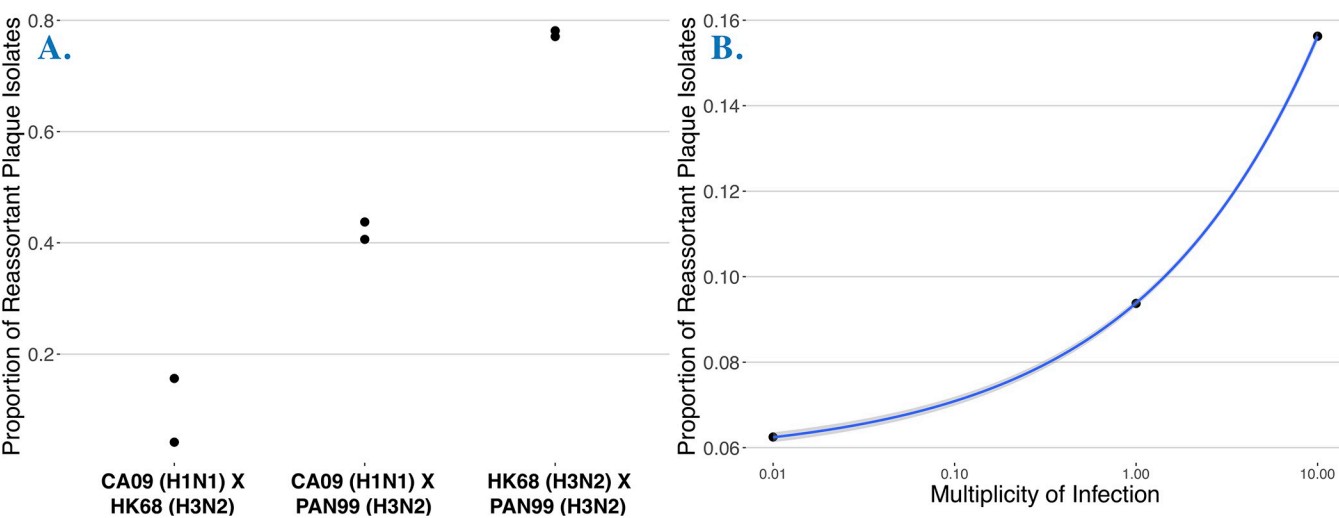

**Fig 2. Genotype by Barcode Sequencing is replicable and measures reassortment accurately. A.** The proportion of reassortant plaque isolates in biological replicates of experimental coinfections initiated at MOI 10. The insets show 8 segment genotype composition of isolated plaques in these replicates representing coinfections with low, medium, and high rates of reassortment. **B.** The proportion of reassortant plaque isolates from experimental coinfections at three multiplicities (0.01, 1, 10) of infection for the CA09 x HK68 experimental coinfection. The gray shading around the blue line represents the standard error around an exponential distribution.

## Reassortment is an emergent property of a pair of strains

We set out to investigate reassortment potential among five human influenza A strains (CA09, HK68, PAN99, SI86, TX12 see **Table 1**) encompassing two pandemics (swine and avian origin), three specific epidemics, and both circulating human subtypes. We conducted all possible pairwise experimental coinfections between these strains (n = 10) in MDCK cells at an MOI of 10 and determined the genotypes of progeny isolated via plaque assay. Genotypes for individual progeny are depicted as rows in each panel (**Fig 3A**), with columns indicating the genotype (i.e. parental origin) for each of the eight segments.

From this primary data, we initially measured and analyzed reassortment frequency, defined as the proportion of progeny harboring at least one segment from a different parent. All pairwise experimental coinfections (n = 10) had measurable reassortment frequencies (**Fig 3A**). Reassortment frequency estimates did not differ significantly (pairwise proportion tests, p value range = 0.132–1.000) when plaque isolates with incomplete genotypes were included or excluded. Thus, reassortment frequencies are presented using all data (i.e. including incomplete genotypes), which is a conservative approach because inclusion of genotypes that are missing segments decreases the chance of detecting a reassortant. Reassortment frequency ranged from 92.71% to 4.17% with an average of 48.95% ± 30.08 (mean ± sd). Overall, 7/10 coinfections generated reassortment frequencies of >40% (**Fig 3B**). Under the assumption of completely free reassortment, the percentage of reassortants expected is 99.22%, i.e. 254/256. One experimental coinfection approached this expectation numerically, CA09xSI86 (92.71%), but like all other coinfections, random expectations for reassortment frequency were rejected (Exact binomial: p < 0.001).

To test if the reassortment frequency calculated for each experimental coinfection (represented by gray bars in **Fig 3B**) differed statistically, we determined whether reassortment frequencies were equal. A test for equal proportions suggested that reassortment frequencies (e.g. values depicted in bars in **Fig 3B**) for experimental coinfections were not equal (p < 0.001). Specifically, a pairwise comparison of proportions revealed 29/45 of the pairwise frequency comparisons were statistically significantly different (Holm adjusted p < 0.05). For instance,

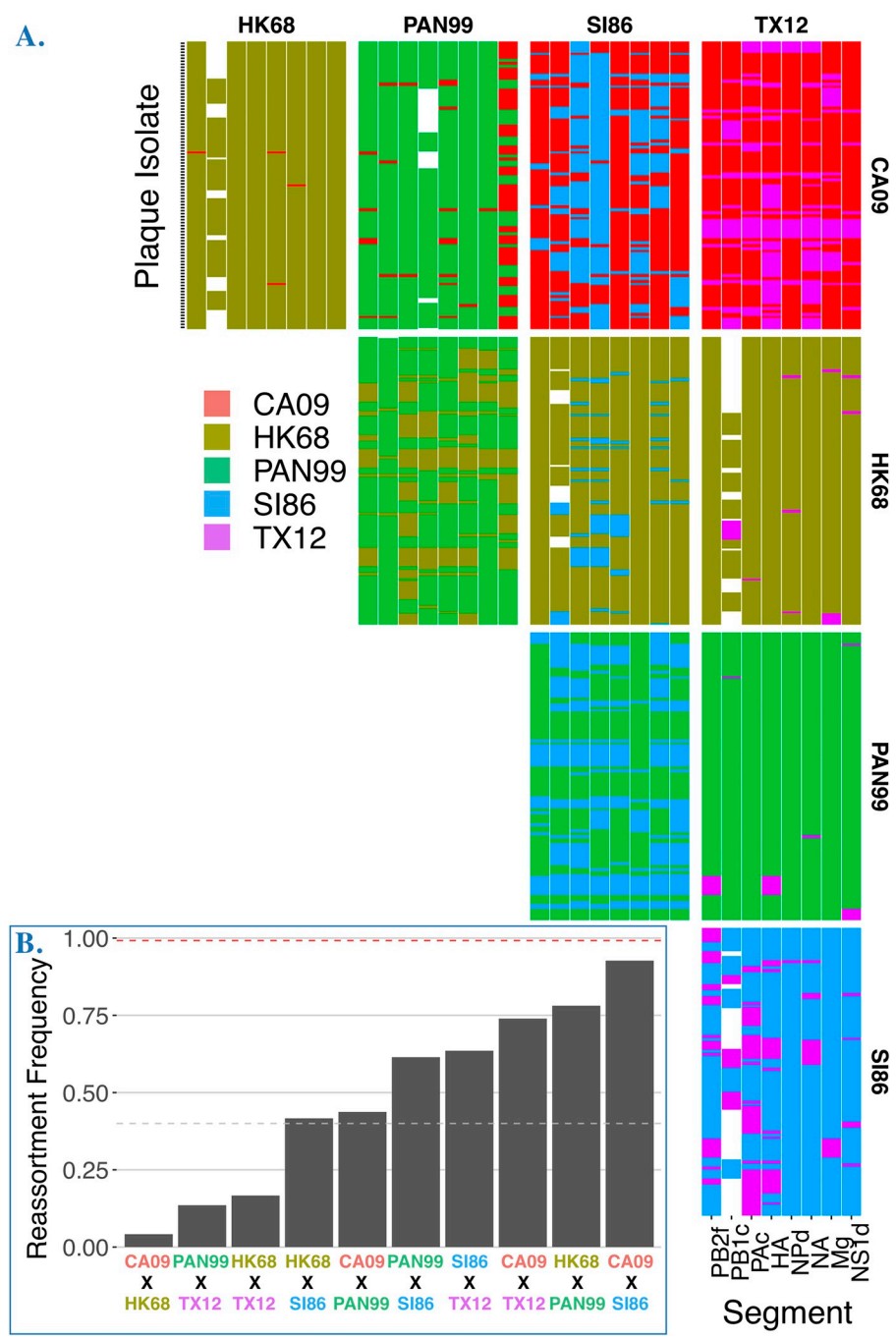

**Fig 3. Reassortment plots and rates determined from plaque isolates. A.** Reassortment plot of genotyped progeny isolated from each coinfection performed in this study. Genotypes for individual progeny are depicted as rows in each panel with columns indicating the genotype (i.e. parental origin) for each of the eight segments. White spaces show genotypes where one or more segments were not identified or recovered. **B.** Reassortment frequencies (proportion of reassortant plaque isolates) for each coinfection. Red line indicates theoretical maximum reassortment, gray line indicates 40% reassortant progeny.

reassortment frequency between the SI86xTX12 (0.635) and PAN99xSI86 (0.615) coinfections were not statistically different, but both were different from CA09xSI86 (0.927). These differences were not explained by genetic similarity (Adj. $R^2$ = -0.07118, p = 0.5438; **Table 2**,

**Table 2. Pairwise percent nucleotide identity among Influenza A strains used in experimental coinfections.** Values above the diagonal exclude antigenic segments HA and NA which are more highly variable than the internal segments.

| Strain | CA09 (H1N1) | HK68 (H3N2) | PAN99 (H3N2) | SI86 (H1N1) | TX12 (H3N2) |
|---|---|---|---|---|---|
| CA09 (H1N1) |  | 86.6 | 85.6 | 84.2 | 85.0 |
| HK68 (H3N2) | 78.1 |  | 94.2 | 92.1 | 92.8 |
| PAN99 (H3N2) | 77.6 | 93.1 |  | 89.3 | 97.0 |
| SI86 (H1N1) | 82.8 | 82.1 | 80.4 |  | 88.4 |
| TX12 (H3N2) | 77.0 | 91.3 | 96.4 | 81.0 |  |

Table A and Fig A in **S1 Text**) or subtype (**Fig 4**, below), suggesting a distinct reassortment frequency emerged from each strain combination (**Fig 3A**).

## Inter- and intrasubtype coinfections do not differ in reassortment frequency

Experimental [19–21] and epidemiological [49] evidence has led to the idea that the subtype of coinfecting influenza A strains can influence reassortment, and specifically that intrasubtype

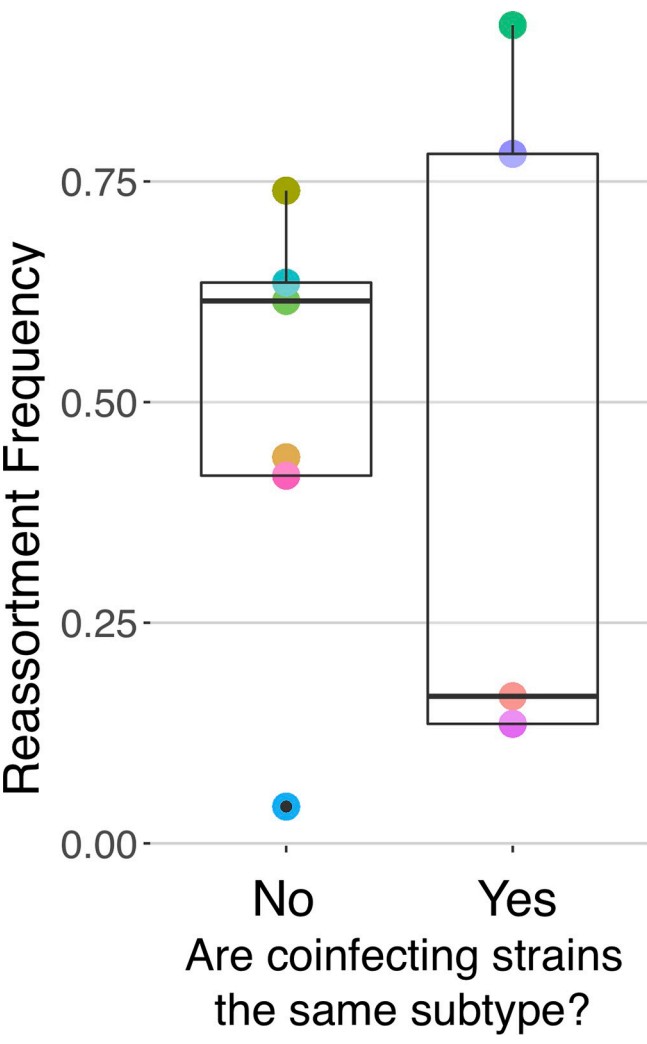

**Fig 4. There is no significant difference in reassortment frequencies in intrasubtype (H1N1 or H3N2) coinfections versus intersubtype coinfections (H1N1xH3N2).**

reassortment should be more common [10,50]. To determine whether antigenic subtype influenced reassortment frequency, we examined reassortment frequencies within and between H3N2 and H1N1 subtypes. The mean intrasubtype reassortment frequency was 0.50 ± 0.410 and the mean for coinfections between different subtypes (i.e. H3N2/H1N1) was 0.48 ± 0.248. There was no statistically significant difference in mean reassortment frequency (t = 0.095, p = 0.9285) of inter- vs. intrasubtype coinfections (**Fig 4**).

## Reassortment frequencies are strain-dependent

Given that reassortment frequency appeared to be an emergent property of each strain combination and did not relate to genetic similarity (**Table 2**) or subtype (**Fig 4**), we set out to determine whether individual strains could influence reassortment frequency of a coinfection and thus represent a trait of individual viruses, as observed for recombination rate in animals [32–34]. Strains generated a range of reassortment frequencies (**Fig 3**) and we were further interested in testing whether each strain consistently biased reassortment frequency (upwards or downwards), independent of the coinfecting partner. We examined the average proportion of reassortants in each experimental coinfection (**Fig 5A**) using ANOVA removing the intercept from the regression, reflecting the biological reality that there is no reassortment if there is no coinfection partner. This analysis showed individual strains were a statistically significant predictor of reassortment frequency (ANOVA: F = 15.75, df = 5, p = 0.004). Model coefficients (**Fig 5B**) represent a statistical calculation of the average individual contribution to reassortment frequency, in the theoretical absence of another strain. Specifically, the analysis suggested PAN99, HK68, and SI86 strains tended to, on average, increase reassortment, regardless of which other strain was coinfecting. For instance, SI86 had a significant ANOVA coefficient of 0.345; across the dataset the average reassortment frequency for coinfections including SI86 was 0.648, with no coinfection below 0.416. In contrast, CA09 did not have a significant coefficient and exhibited the widest range of reassortment frequencies (0.0412–0.927), indicating it did not influence the reassortment consistently. These results suggest that particular strains can significantly push reassortment frequency up or down. Thus, reassortment frequency is an

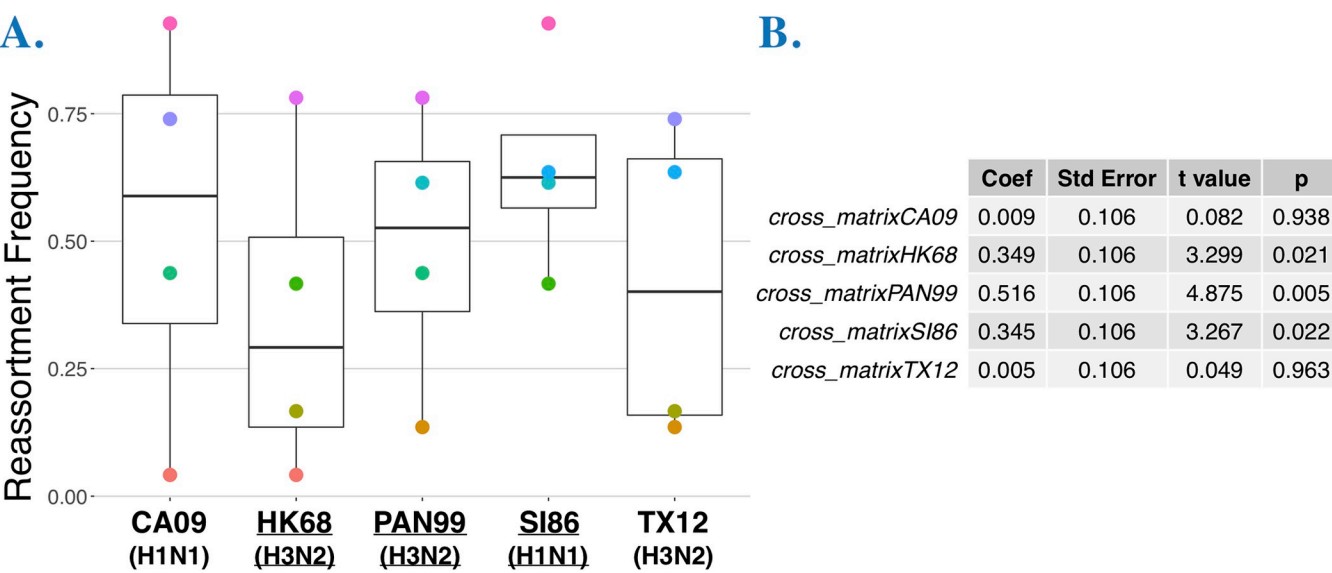

**Fig 5. Reassortment frequency by strain. A.** Reassortment frequencies by strain, calculated from genotyping plaque isolates in ten experimental coinfections. Because reassortment frequency is pairwise property the same reassortment frequencies (points) appear in multiple strains, this is indicated by a shared color. **B**. Coefficients of an ANOVA model, showing that strain is a significant predictor of reassortment frequency, with coefficients indicating individual strain contribution to reassortment frequency.

emergent property of the strain combination but can also be biased towards more or less reassortment by particular strains.

## Viral yield affects the total number of reassortant progeny produced

Multiple studies suggest that cellular coinfection can change influenza A virus production in single-strain infections [51–54], a phenomenon termed multiplicity dependence [52]. We reasoned that the mixed strain coinfections we conducted would affect viral reproduction of each coinfection. Thus, to gain insight into whether viral infection kinetics can affect the *total number* of reassortant progeny, we measured supernatant titers of each experimental coinfection. The titer of each coinfection (**Fig 6A**, *top bars*) was not correlated with reassortment frequency (**Fig 6B**). To estimate the total number of reassortant progeny, we multiplied the titer by the reassortment frequency for each coinfection (**Fig 6A**, *bottom*) and show that the rank order of reassortment is different depending on whether total reassortants or reassortment frequency is used.

## Many segments contribute to reassortment and individually deviate from random representation in progeny with respect to strain

Because reassortant progeny are defined as having one or more genome segments from different parental viruses, reassortment frequencies are not necessarily informative about patterns

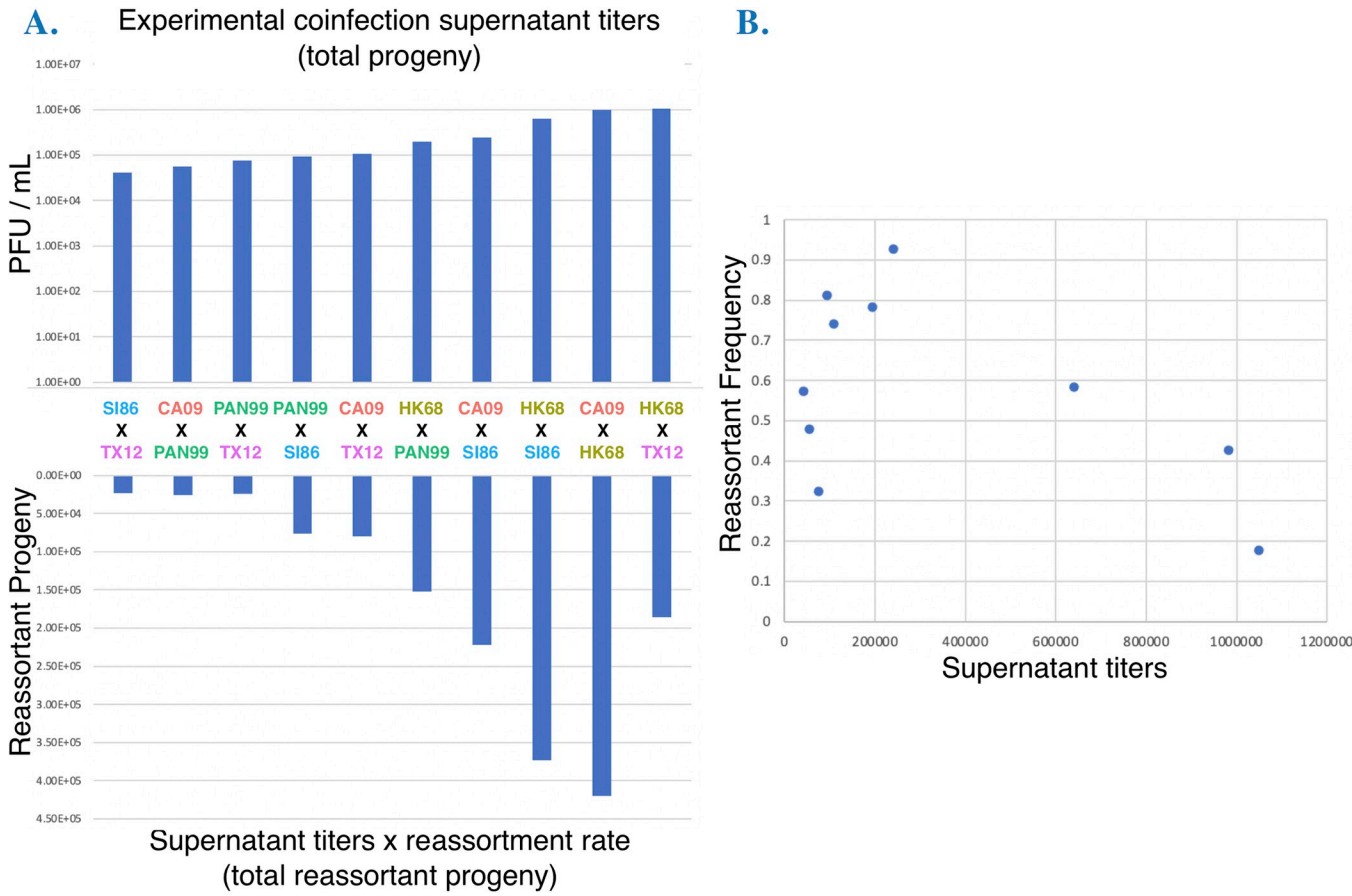

**Fig 6. Viral kinetics of coinfections affect the total production of reassortants. A.** Top bars show final infectious titers (PFU/mL) of each coinfection, in ascending order. Bottom bars show an estimate of the total reassortants produced by the coinfection, by multiplying the corresponding reassortment frequency for each coinfection with its titer. Note that the rank order of total progeny (top bars), and total reassortant progeny (bottom bars), and reassortment frequency (Fig 3B) is different. **B.** Reassortment frequency is not correlated with titers of the same experimental coinfection.

of segment exchange. To gain a segment-centric view on reassortment, we examined genotype representation across all progeny isolated from an experimental coinfection. Under unbiased coinfection conditions, for each of the eight IAV genome segments there is a 50/50 chance of either of the two parental segments being incorporated into a progeny virion, via random assortment. Biases in the parental strain origin of each segment, or deviations from random expectations, indicate potential differences in the relative fitness of segments during incorporation into progeny virions and coinfection. Upon examining the progeny across all coinfections, we first found that, on average, 7.100 ± 1.595 out of the eight segments were involved in reassortment, i.e. each strain contributed at least one of its segment alleles among all isolated plaques (Fig 7A). However, segment alleles were usually statistically biased towards one parental strain (Fig 7B, binomial exact test for deviation from 0.5 proportion at p < 0.05). Across all coinfections, only 13/80 segments conformed to random assortment (Fig 7B; Table B in S1 Text). One intersubtype coinfection, SI86xPAN99, accounted for six of these freely assorting segments, with 6/8 segments adhering to random assortment in this cross. There was no relationship between reassortment frequency and the number of segments with parental strain representation bias (Fig B in S1 Text). Analyzing only reassortant plaques did not qualitatively change the results (Fig C in S1 Text).

## Non-random pairwise segment associations were not limited to divergent strains and correlated with reassortment frequency

Under completely free reassortment, segments randomly assort. Non-random pairwise associations between segments (i.e. linkage) have been observed, particularly for divergent strains. Using 2 influenza strains in a coinfection, there are 4 possibilities for pairwise segment combinations with each expected to be represented in 25% of progeny if there is no bias in pairwise segment associations (for example, in a coinfection between HK68 and SI86 the HA and NA segments could assort in the following ways: $HA_{HK68} + NA_{HK68}$, $HA_{HK68} + NA_{SI86}$, $HA_{SI86} + NA_{HK68}$, $HA_{SI86} + NA_{SI86}$). Two of these combinations have both segments derived from the same parent (i.e. homologous) and the remaining two have segments derived from each parent (i.e. heterologous). The latter is diversity generating, representing new combinations of segments. To simplify pairwise data and examine broad reassortment potential, we focused our analysis on quantifying the heterologous versus homologous plaques in each pairwise segment combination (out of 28 possible with 8 segments). Overall, the average proportion of plaques with heterologous pairwise combinations was significantly different according to the strain combination in each experimental coinfection (Fig 8A, ANOVA F = 14.698, p < 0.0001). This measure did not vary according to the strain similarity or subtype of each coinfection. Reassortment frequency was correlated to the average proportion of heterologous plaques (Adj. $R^2$: 0.505, p = 0.0128; Fig 8B).

Homologous combinations in each locus pair composed most progeny genotypes. Overall, across all coinfections, pairwise loci combinations were represented by 53.68% homologous plaques and 15.96% heterologous plaques (see also Fig 8A). This dominance of parental segment associations led to the question of whether this pattern was driven by singly infected cells–which should be very rare under MOI 10, equal proportion infection conditions–or double infections of only parentals. To examine only plaques known to be derived from cells coinfected by different strains, we examined only plaques with reassortant genotypes, thereby discarding progeny produced from potentially singly-infected cells. Considering only reassortant plaques, we still found that homologous pairwise loci combinations dominated among plaques: 42.28% homologous plaques versus 27.21% heterologous plaques. Differences among the strain combinations in experimental coinfections remained (ANOVA: F = 24.269, p < 0.0001) and did not relate to genetic similarity or subtype.

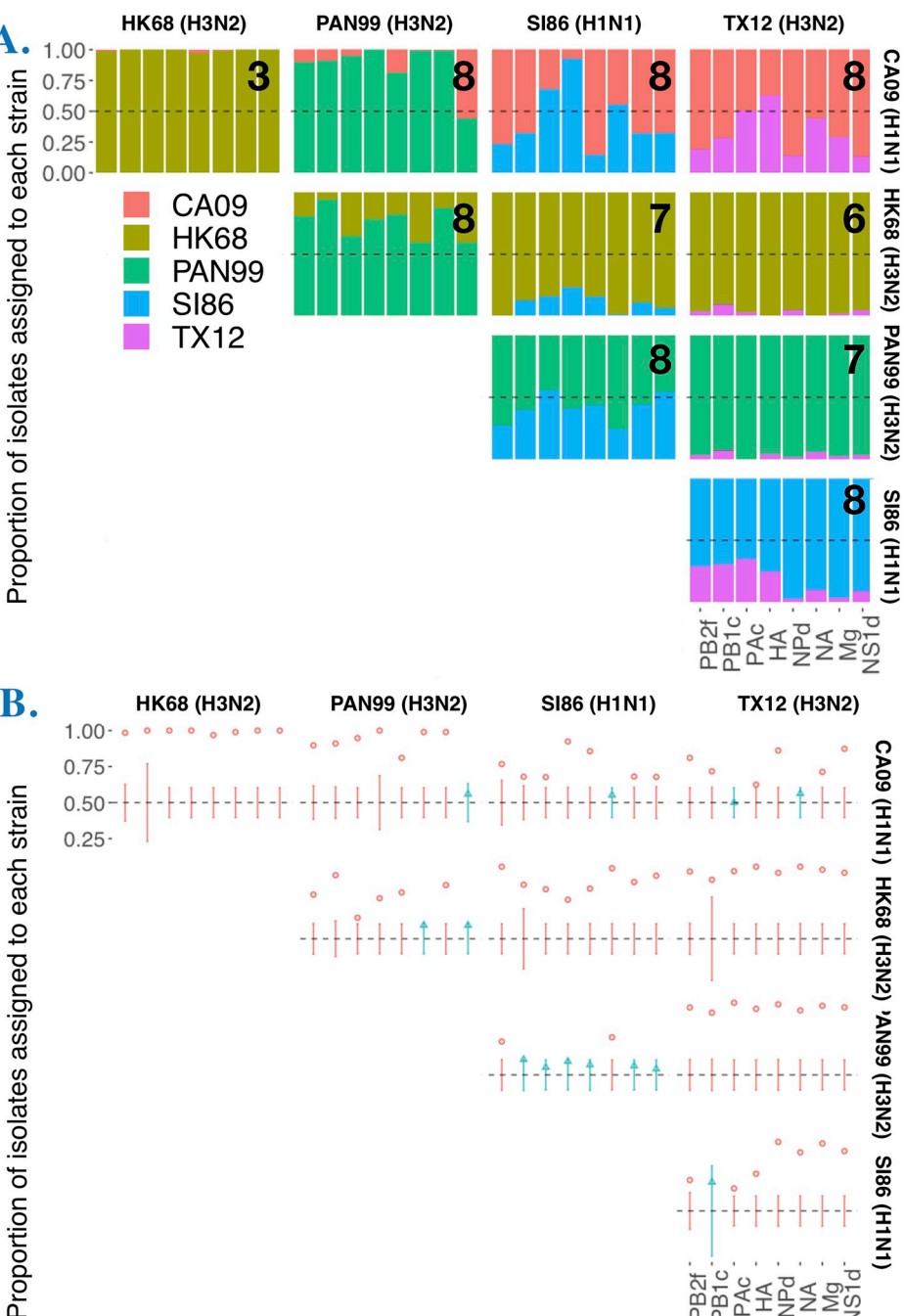

**Fig 7. Many segments participate in genetic exchange and are non-randomly distributed in progeny plaque isolates with regard to strain. A.** Plot shows frequency of each strain's allele for each segment in plaque isolates. The number on the top right corner of each plot indicates the number of segments that participated in reassortment. **B.** Depicts which segment frequencies are within (blue points) or outside (red points) the confidence interval for 50:50 distribution of strain alleles among plaque isolates.

## High reassortment coinfections produce a greater number of unique genotypes

Finally, to examine complete genotypes produced by each experimental coinfection, we determined the eight-segment composition of each plaque and calculated genotype richness (S), the number of unique eight-segment combinations (**Fig 9A**). Experimental coinfections yielded a maximum of 36

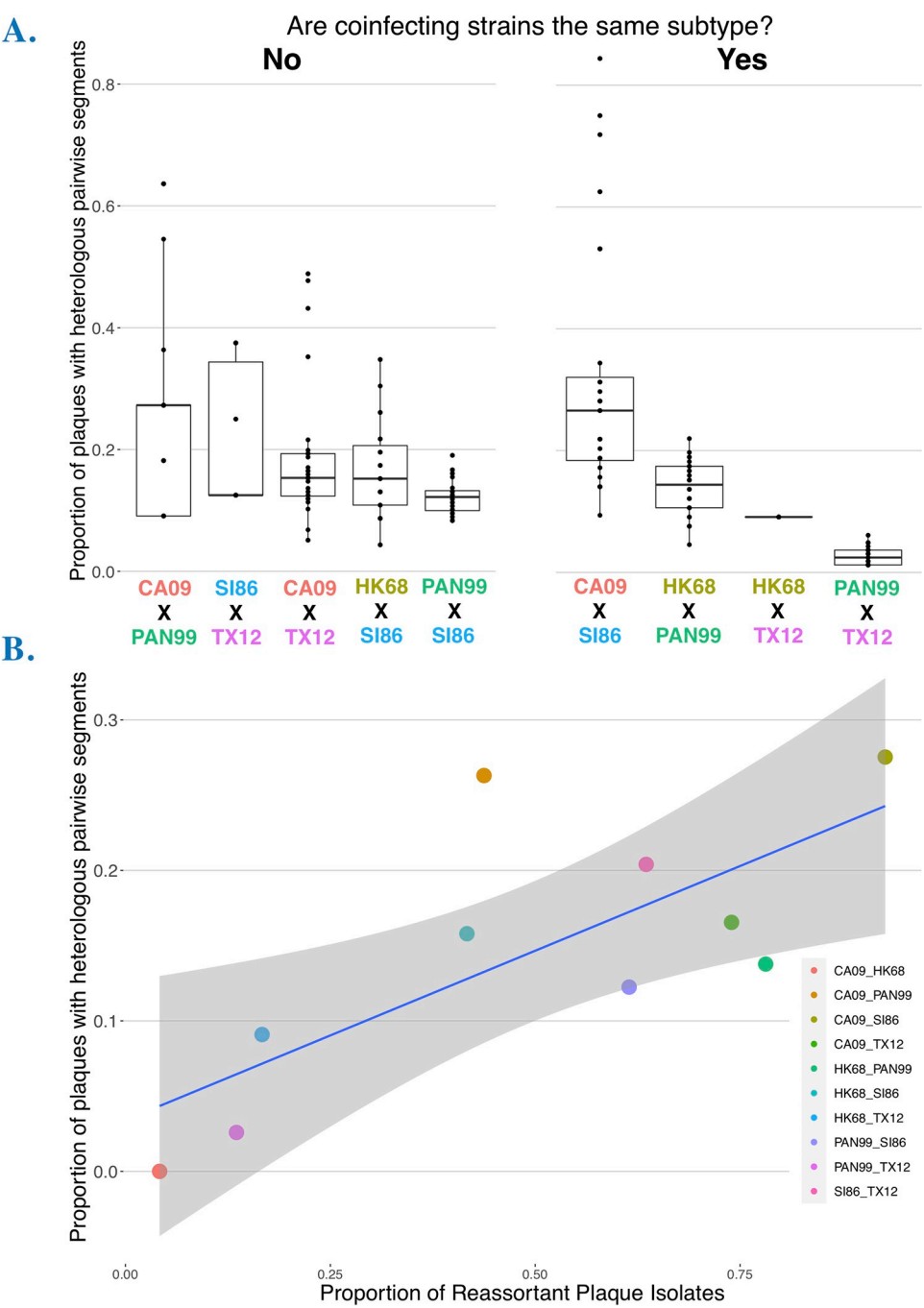

**Fig 8. Pairwise segment combinations across experimental coinfections. A.** Proportion of plaque isolates that had heterologous (i.e. reassortant) genotypes for each pairwise segment combination, represented by points. The left panel shows intersubtype experimental coinfections (i.e. H1N1xH3N2) and the right panel shows intrasubtype (i.e. H1N1xH1N1 or H3N2xH3N2). **B.** Relationship between reassortment frequency (x-axis) and the average proportion of plaque isolates with heterologous genotypes (across all pairwise segment combinations for each experimental coinfection). Each point represents an experimental coinfection.

unique genotypes (range: 1–36) from 12–88 isolated plaques with complete eight-segment genotypes (progeny with missing segment data were excluded). The frequency of reassortment showed a positive correlation with genotype richness (**Fig 9B**, Multiple $R^2$ = 0.8366, p < 0.001).

Because only 94 potential plaques were sampled among $2^8$ = 256 possible combinations (and only a subset of those yielded complete genotypes) we explored the effect of sampling (i.e. how many progeny are genotyped) using a simulation which assumed perfectly random assortment (Bernoulli trial, i.e. each progeny isolate has a 0.5 probability of receiving either parental genotype, independently for each segment). One thousand trials of this simulation suggested that sampling 96 plaques would at best yield 90 unique genotypes (mean ± SD = 80.30 ± 3.05). The coinfection that proportionally generated the most unique genotypes, CA09xSI86 (**Fig 10A**, *red point*), yielded 32 plaques (y-axis) with complete genotypes, of which 20 were unique (x-axis). This yield falls outside of the range of 27–32 unique genotypes predicted (blue lines) in a simulation of free assortment for a sample size of 32 plaques. Thus, increased sampling of progeny is necessary to capture rare genotypes: under free reassortment, even 96 sampled progeny would capture ~31.21% of possible 8-segment genotypes (**Fig 10A**)

To achieve sampling all 256 genotypes under random assortment (**Fig 10A,** *dashed red line*), an average of 1,557 ± 333 progeny would have to be sampled and genotyped (n = 100 trials, blue lines in **Fig 10A**). However, this result is only expected if the coinfecting strains have free assortment between all segments, which is not the case (**Figs 6, 7**). Therefore, to inform on the effect of sampling on estimates of genotype richness and construct a similar genotype accumulation curve, we used biological replicates on a subset of strains. For two replicate HK68xPAN99 coinfections (**Fig 10B**), we screened 188 plaques and obtained a total of 96 complete eight-segment genotypes (x-axis) that yielded 40 unique genotypes (y-axis). If we assume this baseline is the entire genotype space, we can calculate how close each replicate coinfection individually came to capturing all genotypes. In two replicate coinfections, we obtained 77.50% (n = 66 complete genotypes) and 47.50% (n = 30 complete genotypes) of assumed possible eight-segment combinations (**Fig 10B**, *red points*). Similarly, for three replicate CA09x-PAN99 coinfections (**Fig 10C**), we screened 282 plaques and obtained a total of 132 complete eight-segment genotypes that yielded only 14 unique genotypes. In our three replicate coinfections we obtained 35.71% (n = 11), 42.86% (n = 47), 78.57% (n = 74) of assumed possible eight-segment combinations (**Fig 10C**, *red points*). These results suggest that obtaining roughly >60 complete genotypes will recover the majority of genotypes in strain combinations that do not reassort freely. For strain combinations that reassort close to freely, robust sample sizes are required to capture all genotypes because of significant sampling variability due purely to chance.

## Discussion

To quantify the outcomes of coinfection between intact strains and their reassortment potential, we employed a high throughput protocol that can examine diverse human seasonal strains spanning decades. Using this robust data set, we were able to draw conclusions regarding strain traits that may influence reassortment potential, and to examine associations between coinfection outcomes that affect reassortment potential. First, we found that reassortment frequency (proportion of reassortants) is an emergent property of a pair of strains and is not explained by strain similarity or shared antigenic subtype. However, we show evidence that strain identity could serve as a predictor as particular strains tend to favor more or less reassortment, independently of the coinfecting partner. Second, we found that most strain combinations involved all 8 segments in reassortment and that random assortment of individual segments among progeny was rare but was not precluded by intersubtype coinfection. Third, we found evidence of non-random pairwise segment associations. Specifically, most strain combinations yielded homologous linkages but this outcome again was not related to strain

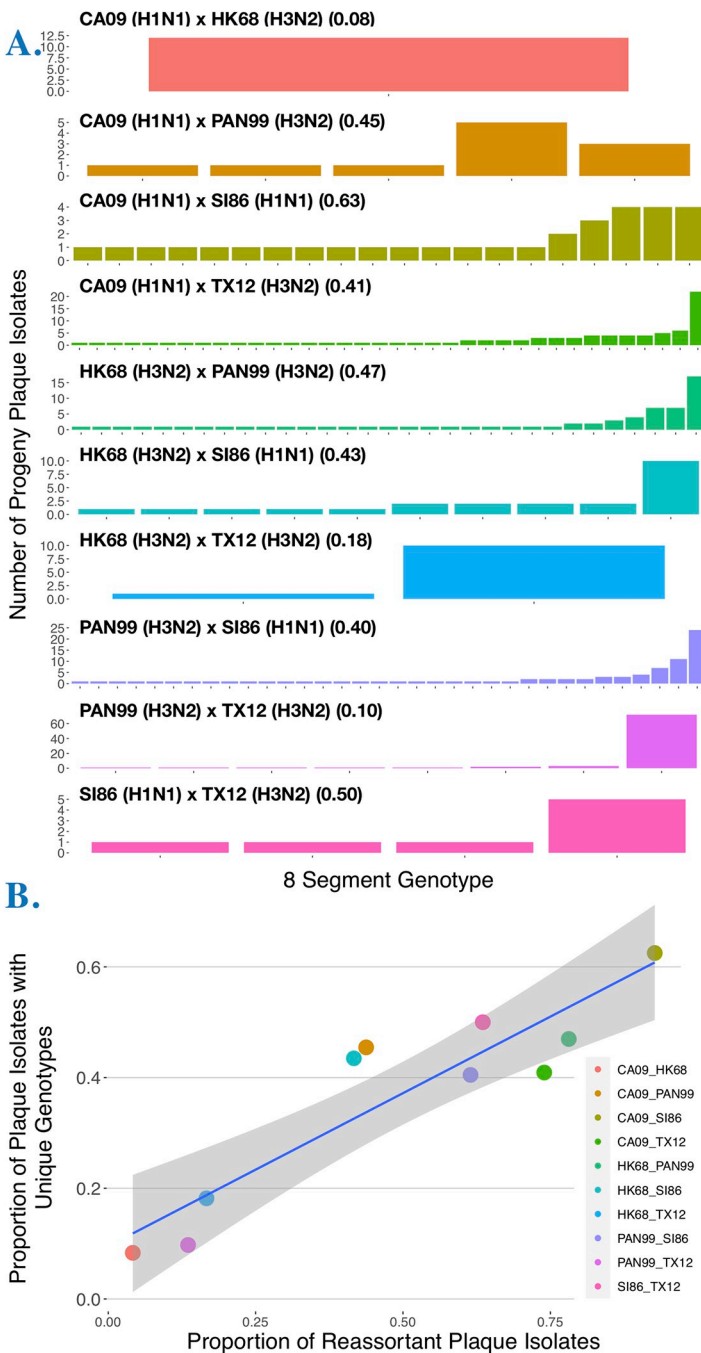

**Fig 9. Experimental coinfections generate varying numbers of unique genotypes. A.** The number of plaques having different 8 segment genotypes are depicted for each experimental coinfection. The reassortment frequency is noted in parentheses next to the strains involved in the coinfection. **B.** Reassortment frequency is positively correlated to the proportion of unique genotypes.

similarity or shared subtype. Fourth, the number of unique genotypes generated by each coinfection spanned a wide range (1–36), again suggesting emergent outcomes from distinct strain combinations. Examining associations between coinfection outcomes, we observed that reassortment frequency is closely correlated to both the proportion of unique genotypes and the

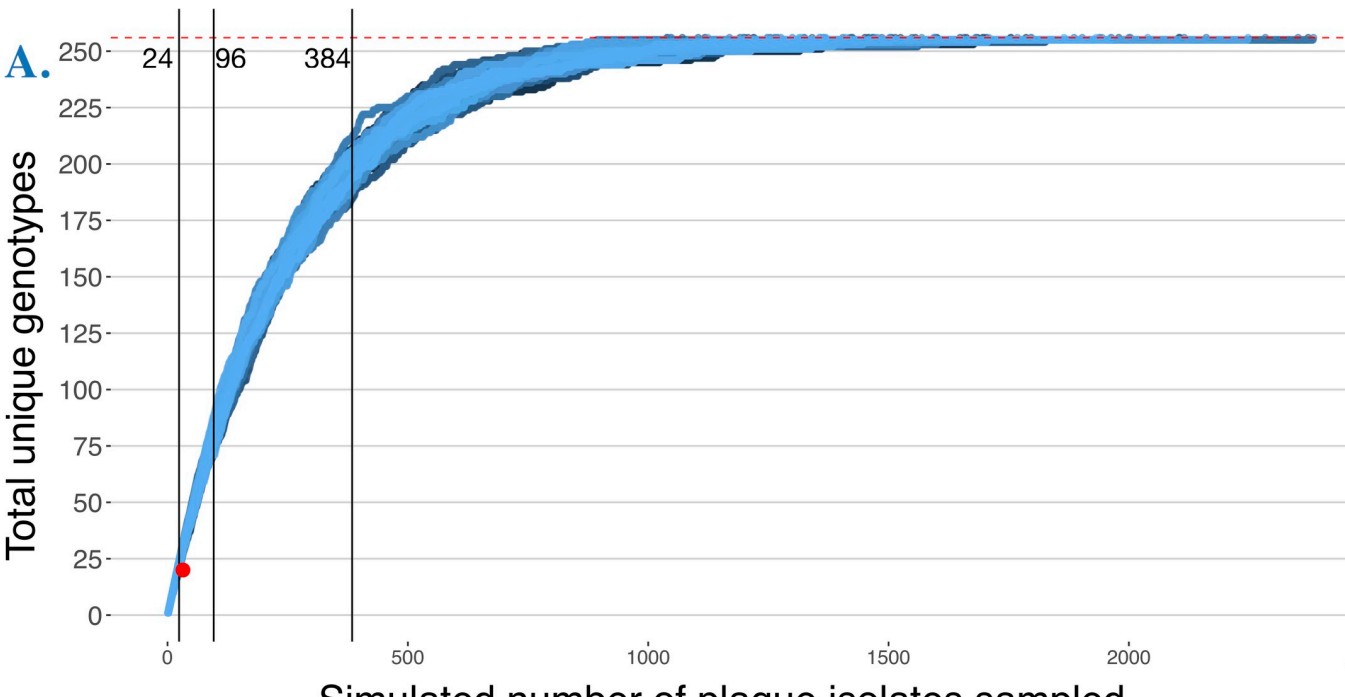

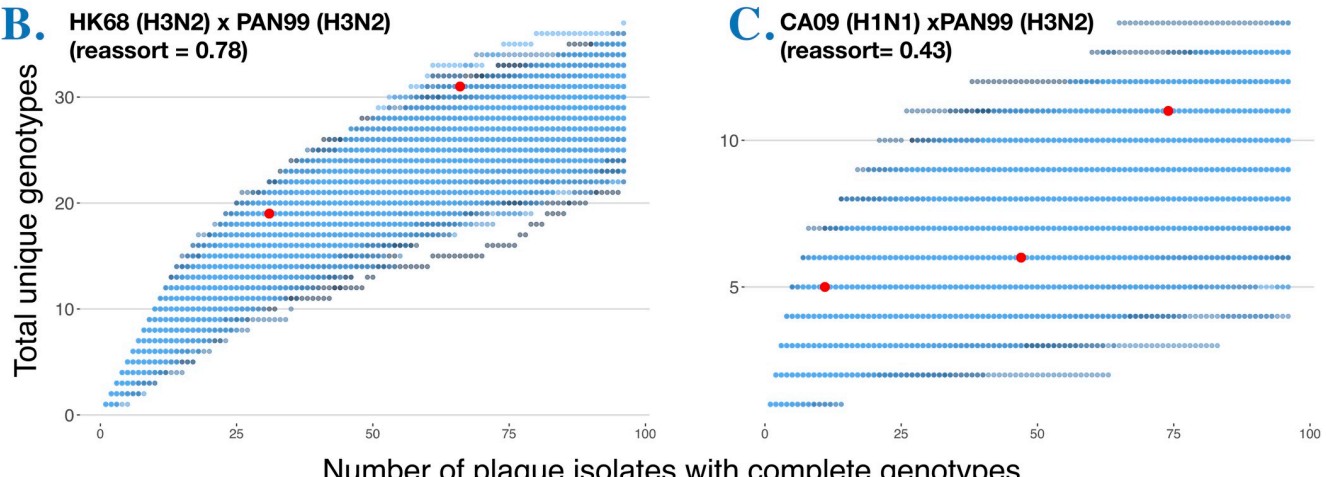

**Fig 10. A large sample of progeny plaque isolates is essential to capture all potential genotypes generated by coinfections. A.** Simulated number of progeny plaque isolates (x-axis) needed to reach 256 genotypes under free reassortment (horizontal red dashed line). Vertical gray lines denote typical sample sizes for molecular biology studies (24, 96, 384). **B.** Replicate coinfection (n = 2, left; n = 3 right) experiments were used as starting data to generate a sample accumulation curve similar to A, but sampling empirical data. Number of unique genotypes found in individual replicate experiments are shown in red, to depict the effect of reduced sample sizes in capturing genotype richness of experimental coinfections.

proportion of progeny with heterologous pairwise segment combinations, but not to segment assortment. Finally, we show that viral kinetics vary independently of reassortment frequency, affecting the total number of reassortants produced by each coinfection.

To date, the highest reported reassortment frequency between human influenza A strains was an average of 88.4% between nearly identical PAN99 strains [25]. In comparison our data

yielded a 92.71% frequency for a CA09xSI86 coinfection (despite the modest sample size, see **Fig 9**), showing that some divergent strains can potentially generate higher reassortment frequencies than near-identical strains, albeit, still short of theoretical free reassortment 99.22% = 254/256. This result suggests that strain similarity is not a prerequisite for a high reassortment frequency between human influenza strains, as first reported by Phipps et al [29] for heterosubtypic coinfection of H3N2 (PAN99) and H1N1 (A/Netherlands/602/2009(H1N1)-like) strains. Our data further support a reversal of the consensus arising from initial experimental coinfection and vaccine production studies, that coinfection of dissimilar strains or subtype mismatched strains limit the production of reassortant progeny [19–21,55].

Thus, overall genetic similarity and subtype do not appear to be consistent predictors of reassortment frequency. By testing diverse human seasonal strains, we were able to determine that reassortment frequency appears to be an emergent property of strain combinations. The question was which strain properties are predictive of reassortment frequency. We present initial evidence that strain identity can serve as a predictor of reassortment frequency, suggesting that genetic exchange may be an individual trait of influenza A strains, analogous to variation among individuals in meiotic recombination between animals [32–34]. Although genetic exchange clearly requires at least two organisms, these may have mechanisms to predispose to more or less genetic exchange, as characterized in mammals (humans, cattle, and sheep) and *Drosophila* [33]. The potential mechanisms that could mediate this trait are plentiful and beyond the scope of this study. Future studies are needed to examine which genotypes and molecular mechanisms underlie the tendency of strains that tend to increase or decrease reassortment.

We found no difference in reassortment frequencies between intrasubtype and intersubtype strain pairings. Influenza A H3N2 and H1N1 subtypes have co-circulated in the human population for nearly half a century, with H1N1 1918 pandemic lineage strains since 1977 [56] and 2009 pandemic lineage strains (H1N1pdm) since that year. Given their co-circulation, it may be expected that H1N2 reassortants would be detected. Indeed, H1N2 reassortants have been identified [10,13,57] and have given rise to epidemics [12] or become epidemiologically significant [9]. However, H1N2 strains have not managed to establish themselves in the human population [12,57]. Our results imply that the barrier for *generating* H1N2 reassortants might be low and thus, given that some H1N2 strains have generated epidemics, the reason why they have not become established likely relates to competition between subtypes in circulation, and not to constraints on reassortment. One limitation of our study is that we could not statistically determine if *individual* subtypes differed in reassortment frequency (i.e. H1N1/H1N1 versus H3N2/H3N2 coinfections), due to sample size constraints. This may be important data to forecast the risk of immune evasion due to intrasubtypic reassortment in seasons where H1N1 or H3N2 subtypes dominate.

Even in coinfections with the highest reassortment frequencies, segment exchange was not free. Overall, there was a bias towards specific parental segments among the progeny homologous pairwise combinations in reassortants. This finding is in line with numerous experimental coinfection studies, dating from the first study of this type [18–21,29]. Our novel finding is that the pattern of non-random segment exchange is not mediated by the genetic distance of strains involved or intersubtypic combinations, as widespread consensus in the literature held [18,19,21]. One of the strains that showed the freest segment assortment among progeny (**Fig 6B**: 6/8 segments assorting randomly) was SI86xPAN99, an intersubtype coinfection. Likewise, the strain with the most heterologous pairwise segment combinations was SI86xCA09, from divergent H1N1 lineages. These findings within human strains echo other studies that showed robust, if not random, segment exchange among distantly related strains [23,58].

Although we did not find strain combinations that approached random assortment between segments, we did find that there was a tight and statistically significant correlation between reassortment frequency and both the proportion of unique genotypes and the proportion of heterologous pairwise segment combinations (these two are most likely tightly correlated). While *prima fasciae* this seems like a trivial observation, there is no *a priori* reason why these should be linked; for example, in the extreme case of a single favorable heterologous pairing of two segments, there could be only one genotype produced and thus 100% reassortant progeny. If the above correlation holds, the implication is that strain combinations that yield a higher frequency of reassortants also produce more diverse combinations.

Another important relationship between coinfection outcomes we found is that viral kinetics vary independent of reassortment frequency, affecting the total number of reassortants produced by each coinfection. Reassortment studies typically focus on rates and proportions among progeny. However, an increasing number of studies show that coinfection itself can affect influenza A virus growth kinetics [51–53,59], a long-held observation from vaccine production studies [54,60]. In single strain infections this phenomenon has been termed multiplicity dependence [52]. We therefore reasoned that mixed coinfection (at our high MOI experimental conditions) could impact viral production. We find that highly productive coinfections do not have a higher frequencies of reassortment (and vice versa), but productivity will have an impact on the total number of reassortants in all cases. The lack of correlation between viral production and reassortment frequency could assist in public health risk assessment by using these traits as separate criteria when assessing reassortment potential.

A last technical observation is that data arising from simulations and empirical data suggest that it is necessary to sample a large number of progeny plaque isolates to cover the potential diversity generated by each coinfection and to avoid random sampling variability (due purely to chance). This observation is purely statistical and this limitation applies to the current study. However, provided complete genotypes are obtained, 96 samples will provide most unique genotypes, at least in strain combinations that do not have completely free segment exchange.

Our study design has implications for data analysis and interpretation of all studies examining reassortment: we sought to examine the outcomes of coinfection of intact strains and any reassortment potential. We focused on maintaining unbiased conditions that examined a single cycle of replication [25]. By design we avoid making inferences regarding the fitness of progeny viruses (beyond capacity to initiate infection in plaque assay). As indicated in the first experimental coinfection study [18], it is important to analyze reassortants arising from a single cycle of replication, as multicycle growth may enrich for particular isolates, due to growth competition among progeny unrelated to the intracellular process of reassortment. Furthermore, when assaying viral "fitness" in the lab, the conclusions are applicable only to those conditions, which may or may not be representative of conditions in nature. A limitation of our study design in determining reassortment potential is that we cannot distinguish between solo infections (which are expected to be rare at MOI = 10) or double infections of the same strain in a cell. Distinguishing those outcomes at the single cell level is technically challenging, and we will embark on this objective in future studies. In particular, one cannot assume that progeny viruses with parental genotypes arise only from cells infected with one strain; they may simply not reassort. Our goal was to examine the overall outcome of coinfection of intact strains to reveal areas for future investigation.

At an evolutionary level, our data suggest that viral genetic exchange is potentially an individual social trait subject to natural selection [61]. Several papers on influenza [26,27,52] and other segmented viruses [30,62,63] have suggested this possibility, and here we provide experimental evidence to establish this conclusion more firmly, suggesting some directions for such

future experiments. Segment mismatch studies should include more segments, as protein compatibility may be strain dependent and protein incompatibilities (e.g. among polymerase complex subunits) can be less severe among divergent strains [23] than more closely related strains [21,64]. At an extreme, a study of influenza A & B reassortment has even shown successful intertypic expression of the HA segment [65]. For studies of RNA-RNA interactions and viral assembly, our data support the idea that RNA-RNA interaction networks among segments are not conserved [66,67], and may retain flexibility for interactions with different strains [68]. Finally, for most studies of coinfection, our data suggest that conclusions based on one or two strains should be extrapolated cautiously. For instance, intersubtype reassortment is often assumed to be less pervasive than intrasubtype reassortment [9] and our data suggests weak evidence for this assumption (**Fig 4**), or at a minimum, that there will be some exceptions.

The evolutionary observation of emergent and strain dependent reassortment also has some important public health implications. First, the tendency of particular strains to increase reassortment frequency (and potentially generate more unique genotypes) implies that the risk of reassortment is not evenly spread among strains. Identification of these high-reassortment strains could improve pandemic preparedness or reveal new treatment targets, particularly if specific loci can be used as markers for increased genetic exchange. One example that has been discussed in the literature is avian H9N2 viruses which tend to provide backbones for avian H7N9s [69,70], creating a highly reassortant platform that is responsible for recent human infections [71,72]. Second, exchange of a subset of segments even at a low frequency (e.g. SI86xHK86, which primarily exchanges HA, PA, and NP segments), could be potentially important in the emergence of novel strains under the correct selective regimes, particularly if those segments carry virulence or host tropism determinants. Thus, we believe for pandemic preparedness our base assumption should be that in virtually every coinfection influenza viruses are shuffling the deck, not perfectly but constantly.

## Supporting information

**S1 Textpdf. Supplementary information including supplementary figures, tables, and detailed methods.**
(DOCX)

## Acknowledgments

Ted Ross kindly provided strains from his collection. Graham Coop and Daniel Runcie provided feedback on specific analyses.

## Author Contributions

**Conceptualization:** Kishana Y. Taylor, A. J. Campbell, Bin Zhou, David Gresham, Elodie Ghedin, Samuel L. Díaz Muñoz.

**Data curation:** Courtney Mattson, Samuel L. Díaz Muñoz.

**Formal analysis:** A. J. Campbell, David Gresham, Elodie Ghedin, Samuel L. Díaz Muñoz.

**Funding acquisition:** David Gresham, Elodie Ghedin, Samuel L. Díaz Muñoz.

**Investigation:** Kishana Y. Taylor, Ilechukwu Agu, Ivy José, Sari Mäntynen, A. J. Campbell, Tsui-Wen Chou, Bin Zhou, David Gresham, Elodie Ghedin, Samuel L. Díaz Muñoz.

**Methodology:** Kishana Y. Taylor, Ilechukwu Agu, Ivy José, Sari Mäntynen, Bin Zhou, David Gresham, Elodie Ghedin, Samuel L. Díaz Muñoz.

**Project administration:** Samuel L. Díaz Muñoz.

**Resources:** Tsui-Wen Chou, Bin Zhou, David Gresham, Elodie Ghedin, Samuel L. Díaz Muñoz.

**Software:** David Gresham, Samuel L. Díaz Muñoz.

**Supervision:** David Gresham, Elodie Ghedin, Samuel L. Díaz Muñoz.

**Validation:** Kishana Y. Taylor, Ilechukwu Agu, Ivy José, Sari Mäntynen, A. J. Campbell, Samuel L. Díaz Muñoz.

**Visualization:** David Gresham, Elodie Ghedin, Samuel L. Díaz Muñoz.

**Writing – original draft:** Samuel L. Díaz Muñoz.

**Writing – review & editing:** Kishana Y. Taylor, A. J. Campbell, David Gresham, Elodie Ghedin, Samuel L. Díaz Muñoz.

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
