## [Decision Letter · Decision Letter 0]

12 Dec 2022

Dear Dr Díaz-Muñoz,

Thank you very much for submitting your manuscript "Influenza A virus reassortment is strain dependent" for consideration at PLOS Pathogens. As with all papers reviewed by the journal, your manuscript was reviewed by members of the editorial board and by several independent reviewers. In light of the reviews (below this email), we would like to invite the resubmission of a significantly-revised version that takes into account the reviewers' comments.

We cannot make any decision about publication until we have seen the revised manuscript and your response to the reviewers' comments. Your revised manuscript is also likely to be sent to reviewers for further evaluation.

Sincerely,

Jacob S. Yount

Academic Editor

PLOS Pathogens

Ana Fernandez-Sesma

Section Editor

PLOS Pathogens

Kasturi Haldar

Editor-in-Chief

PLOS Pathogens

orcid.org/0000-0001-5065-158X

Michael Malim

Editor-in-Chief

PLOS Pathogens

orcid.org/0000-0002-7699-2064

Reviewer's Responses to Questions

**Part I - Summary**

Reviewer #1: In their manuscript, Taylor et. al. describe their study of reassortment of human seasonal influenza strains using in vitro and in silico approaches. Overall, the paper is well written, relatively clear, and the data is robust and novel. The findings are of great interest, especially in regard to pandemic prevention and future events which may result in novel, reassortant strains. However, some issues should be addressed before the paper is acceptable for publication.

1) The authors should spend a bit more time in the introduction discussing the overall risk of coinfection with both a A/H1N1 and A/H3N2 strain within a given season to increase public health relevance.

2) How do the strains utilized in this study differ from their egg passged original stocks? Was sequencing performed both prior and post initial passaging in MDCK? How a=many passages were performed in MDCK before use in their experiments? Was each strain ensured to be of the same passage and same MDCK-associated adaptive mutations before coinfection experimentation began?

3) How was coinfection of single cells detected and confirmed? It would be ideal to show that the MOI utilized resulted in a significant (and equivalent) number of cells was coinfected in each experimental condition.

4) It is highly unclear if infections were performed first them plaque assays were used to isolate individual, viable viral strains or if mock infections were performed and individually infected cells were chosen. The former experiment represents true coinfection and production of single, viable strains while the second just represents mixture within a cell and could result in defective viral particles clinging to the surface of the cell. Please consider describing the experimentation in detail to make clear.

5) 16.3x seems a bit low for coverage of such an important segment as PB1 – what optimization was performed to try to increase this yield, especially for a strain that has important implications for viral replication. This does not appear to be dealt with experimentally in the protocol and just in silico, which is likely insufficient for overall study design.

6) A panel of 5 viruses (3x H3N2 and 2x H1N1) from 2 geographic locations (Asia and North America) does not really constitute as wide of a panel of viruses as implicated in the text, please consider toning down the language and further classifying the full differences between these strains that make them the appropriate choice for these experiments, especially since very few of them would have been cocirculating at any given time. More biologically relevant, currently circulating strains (such as those from before and after COVID-19) may be a better representation of the true risks of this reassortment at the current time.

7) Figure 2, 6A, is extremely hard to interpret in grayscale…please consider making colors a bit more definitive.

8) As mentioned in the results and discussion, the authors mention some issues with undetectable results or insignificance due to sample size constraints. While the calculation how many plaques would be required is interesting (and justifiable for some analyses), the lack of additional data does diminish the final result of the paper and further individual subtype recombination may need to be explored.

9) The discussion about highly productive infection in the discuss may not be justified as they authors do not directly measure virulence or any sort of viral replication as associated with each individual plaque isolate.

10) Do the authors intend to perform these studies using potentially zoonotic strains as well as indicated in the last paragraph of the discussion? This would be extremely biologically relevant, especially with the high frequency of A/H5Nx 2.3.4.4b infections being detected in mammals in Europe and North America as well as the large influenza season currently affecting the northern hemisphere.

11) Was any attempt made to perform these analyses on clinical isolates from true coinfections to correlate the number of reassortants observed with what is observed in vitro/in silico?

Reviewer #2: Taylor et al. establish a genotyping approach and quantify segmental reassortment between five seasonal influenza viruses of H1N1 and H3N2 subtypes collected over 41 years, and identify several interesting details on reassortment. These include (a) some viruses can influence reassortment independent of its co-infecting partner (b) reassortment involves all eight segments with little preference towards specific segments, and (c) linkages between segments were found, but they were not related to sequence similarity or subtype. Overall, this is a nice study that significantly advances our understanding of segmental reassortment. The Results and Figures were presented clearly, but the text could be clearer.

**Part II – Major Issues: Key Experiments Required for Acceptance**

Reviewer #1: Please see Part I above

Reviewer #2: 1. Too much hype is made that the viruses span 41 years, "a diverse panel of human influenza virus strains to encompass 41 years of epidemics, multiple geographic locations, and both circulating human subtypes A/H1N1 and A/H3N2". However, this phrase has some issues.

- The five strains do not encompass epidemics for 41 years. Instead, they represent two pandemics and some other specific epidemics. CA09 originated from swine, and the origin of HK68 is assumed to be birds.

- The mention of geography is jarring because human influenza is a globally circulating virus.

- I can accept that the strains represent a diverse panel of human influenza 'A' viruses.

2. Interestingly, all three strains that tended to increase reassortment were older (PAN99, HK68, and SI86), which likely underwent more passage than recent viruses. Can adaptation to cells influence these findings?

3. Results "Thus, reassortment frequencies are presented using all data, which is a conservative approach because inclusion of genotypes that are missing segments decreases the chance of detecting a reassortant."

Please clarify whether incomplete genomes were included/excluded.

4. Parts of the Discussion that could be easier to read.

Third, .... with homologous linkages preferred"

Fourth, "we found a range" in the number of unique genotypes generated by each coinfection

"An important study design and conceptual element regarding this work bears discussion"

**Part III – Minor Issues: Editorial and Data Presentation Modifications**

Reviewer #1: Please see Part I above

Reviewer #2: Replace "Isolation" with Region/Country in Table 1.

"protocol was independently developed and varies in some experimental steps". Can the author briefly mention these steps in the subsequent lines?

PLOS authors have the option to publish the peer review history of their article (what does this mean?). If published, this will include your full peer review and any attached files.

Reviewer #1: No

Reviewer #2: No
---

## [Editor Report · Decision Letter 1]

26 Jan 2023

Dear Dr Díaz-Muñoz,

We are pleased to inform you that your manuscript 'Influenza A virus reassortment is strain dependent' has been provisionally accepted for publication in PLOS Pathogens.

Best regards,

Jacob S. Yount

Academic Editor

PLOS Pathogens

Ana Fernandez-Sesma

Section Editor

PLOS Pathogens

Kasturi Haldar

Editor-in-Chief

PLOS Pathogens

orcid.org/0000-0001-5065-158X

Michael Malim

Editor-in-Chief

PLOS Pathogens

orcid.org/0000-0002-7699-2064
---

## [Editor Report · Acceptance letter]

24 Feb 2023

Dear Dr Díaz-Muñoz,

We are delighted to inform you that your manuscript, "Influenza A virus reassortment is strain dependent," has been formally accepted for publication in PLOS Pathogens.

Best regards,

Kasturi Haldar

Editor-in-Chief

PLOS Pathogens

orcid.org/0000-0001-5065-158X

Michael Malim

Editor-in-Chief

PLOS Pathogens

orcid.org/0000-0002-7699-2064